# Evaluation of Anti-Methicillin-Resistant *Staphylococcus aureus* (MRSA) Prescribing Habits in Patients with a Positive MRSA Nasal Swab in the Absence of Positive Cultures

**DOI:** 10.3390/pharmacy11030081

**Published:** 2023-05-03

**Authors:** Madeline Pelham, Madeline Ganter, Joshua Eudy, Daniel T. Anderson

**Affiliations:** 1Department of Clinical and Administrative Pharmacy, University of Georgia College of Pharmacy, Augusta, GA 30912, USA; 2Department of Pharmacy, Augusta University Medical Center, Augusta, GA 30912, USA

**Keywords:** methicillin-resistant *Staphylococcus aureus*, nasal swab, pneumonia, antimicrobial stewardship

## Abstract

Methicillin-resistant *Staphylococcus aureus* (MRSA) polymerase chain reaction (PCR) nasal swabs are guideline-recommended de-escalation tools in certain patients with pneumonia. Prior studies have demonstrated reduced anti-MRSA therapy with negative results, but the impact on durations of therapy has been poorly elucidated in patients with positive PCRs. The objective of this review was to evaluate anti-MRSA treatment durations in patients with a positive MRSA PCR in the absence of MRSA growth on culture. This was a single-center, retrospective observational study evaluating 52 hospitalized, adult patients receiving anti-MRSA therapy with positive MRSA PCRs. The overall median duration of anti-MRSA therapy was five days, including a median of four days after PCR results. This was consistent among intensive care unit (ICU) and non-ICU patient populations and in patients with suspected community-acquired pneumonia (CAP). Among patients with hospital-acquired pneumonia (HAP), the median duration of anti-MRSA therapy was seven days, with a median of six days after PCR results. Overall, patients received a median duration of anti-MRSA therapy that would constitute a full treatment course for many respiratory infections, which indicates that providers may equate a positive MRSA nasal PCR with positive culture growth and highlights the need for education on the interpretation of positive tests.

## 1. Introduction

While empiric methicillin-resistant *Staphylococcus aureus* (MRSA) therapy is not recommended for all patients with pneumonia, it may be considered in certain patients or clinical scenarios [1,2]. The MRSA nasal polymerase chain reaction (PCR) test is a guideline-recommended tool that may be used to curb the use of unnecessary anti-MRSA antibiotics.

The MRSA nasal PCR has a high negative predictive value (NPV) of 94.8–99.2% for MRSA pneumonia [3,4,5,6,7]. Studies have demonstrated a significant reduction in durations of anti-MRSA therapy when negative PCR results are used as a de-escalation tool [8,9,10,11,12,13,14]. The use of MRSA PCR screening in pneumonia has also been associated with reduced antibiotic costs [9] and lengths of stay [11]. Based on the high NPV of the test, the American Thoracic Society (ATS) and Infectious Diseases Society of America (IDSA) guidelines for community-acquired pneumonia (CAP) suggest obtaining nasal PCRs in patients with suspected MRSA and withholding or discontinuing MRSA coverage if the result is negative [1].

However, MRSA nasal PCR only has a positive predictive value (PPV) of 30–56.8% [3,4,5,6,7]. Therefore, positive results should be interpreted with caution and are not recommended as a basis to initiate or continue anti-MRSA therapy. The low PPV may result from high rates of MRSA nares colonization, even in the absence of lower respiratory tract infection. For this reason, the IDSA/ATS CAP guidelines recommend obtaining respiratory cultures in all patients with suspected MRSA infection and using culture results to guide anti-MRSA therapy in the event of a positive PCR result [1].

In 2020, our institution implemented a pharmacist-driven MRSA nasal PCR screening protocol when an anti-MRSA agent is ordered for pneumonia. The intent was to facilitate early antibiotic de-escalation based on negative PCR results. Although a positive MRSA PCR is a poor predictor of MRSA infection, we hypothesized that positive MRSA PCR results led to longer anti-MRSA treatment durations. The objective of this study was to assess the length of anti-MRSA therapy in patients with positive MRSA PCRs in the absence of MRSA growth on culture.

## 2. Materials and Methods

### 2.1. Study Design

This was a retrospective, observational study of hospitalized, adult patients at an academic medical center. Patients were included if they had a positive MRSA nasal PCR between 1 March 2022 and 31 August 2022. Patients who received less than 24 h of anti-MRSA therapy, had a length of stay of less than 48 h, had MRSA growth on any respiratory or non-respiratory culture within 90 days prior to collection of an MRSA nasal PCR, or had MRSA growth after a positive PCR result were excluded from the analysis. The institutional protocol for the study site calls for MRSA nasal PCR utilization specifically for suspected pneumonia. However, it is able to be used for other indications at physician discretion, and all patients who otherwise met the inclusion criteria were included for analysis. This study has been reviewed by the Augusta University IRB and is not considered to be human subject research.

### 2.2. Data Collection

A list of patients with positive MRSA nasal PCRs during the study period was screened to identify eligible patients. Data points including patient demographics, type of pneumonia or other infection, duration and timing of anti-MRSA therapy, and respiratory culture data were collected through electronic medical record review.

### 2.3. Definitions

CAP, hospital-acquired pneumonia (HAP), and ventilator associated pneumonia (VAP) were defined in accordance with current IDSA/ATS guidelines [1,2]. Pneumonia was classified as CAP if it developed within 48 h of hospital admission or was present on admission, HAP if it developed >48 h after admission, or VAP if it developed >48 h after intubation. Anti-MRSA agents included in this review include vancomycin, linezolid, daptomycin, ceftaroline, trimethoprim/sulfamethoxazole, and doxycycline. Physician documentation was reviewed to assess the indication for MRSA nasal PCR when it was ordered for indications other than respiratory infections. 

### 2.4. Statistical Analysis

Descriptive statistics were used to analyze the data. Categorical variables are reported as frequencies and percentages, and durations of therapy are reported as medians and interquartile ranges, as these data were not normally distributed.

## 3. Results

### 3.1. Inclusion and Exclusion

One hundred forty-one patients with a positive MRSA nasal PCR during the study period were identified. A total of 89 patients were excluded: 30 were on anti-MRSA therapy for less than 24 h, 3 had a length of stay of less than 48 h, 11 grew MRSA in a culture within 90 days prior to nasal swab collection, 37 grew MRSA in a respiratory culture following a positive PCR result, and 8 grew MRSA in a non-respiratory culture following a positive PCR result. The remaining 52 patients were included in the analysis.

### 3.2. Baseline Characteristics

Patient demographics and respiratory culture data are reported in Table 1. This study assessed an equal number of intensive care unit (ICU) and non-ICU patients. Thirty-seven patients (71.2%) were treated for pneumonia, with CAP being the most common diagnosis (*n* = 28), followed by HAP (*n* = 8) and VAP (*n* = 1). MRSA nasal PCRs were collected for 15 patients receiving anti-MRSA therapy for various non-respiratory infections.

Overall, a respiratory culture was obtained for 24 patients (46.2%). Of the 37 patients with pneumonia, 24 (64.9%) had a respiratory culture obtained; this encompassed 16 (57.1%) patients with CAP, 7 (87.5%) patients with HAP, and 1 (100%) patient with VAP.

### 3.3. Duration of Anti-MRSA Therapy

Across the cohort, the median duration of anti-MRSA therapy was five days (Table 2). The median time from positive MRSA nasal PCR result to discontinuation of anti-MRSA therapy was four days. These results were consistent, regardless of ICU status. The median duration of anti-MRSA therapy was four days for CAP and seven days for HAP.

## 4. Discussion

In this review of 52 patients with a positive MRSA nasal PCR and no MRSA growth on respiratory culture, the median duration of anti-MRSA therapy was four days in patients with CAP and seven days in patients with HAP. These durations closely mirror current guideline-recommended antibiotic durations of therapy, i.e., five days for CAP and seven days for HAP [1,2], indicating that providers may equate a positive MRSA nasal PCR to a positive respiratory culture. There was no difference in durations of anti-MRSA therapy or time to discontinuation of anti-MRSA therapy after positive PCR between ICU and non-ICU patients, indicating that severity of illness did not impact overall prescribing habits.

Respiratory cultures were obtained for only 64.9% of patients with suspected pneumonia and 46.2% of the total cohort. Of note, five (20%) cultures collected in this cohort were sputum cultures. The high potential for contamination of sputum cultures may have influenced provider interpretation of these results. The 2019 ATS/IDSA CAP guidelines do not provide strong recommendations for the use of routine sputum cultures, noting a lack of evidence supporting their use and antimicrobial stewardship concerns with contamination or colonization. Therefore, they abstain from making a recommendation for or against the routine use of sputum cultures for CAP diagnosis. One exception to this is in hospitalized patients receiving anti-MRSA therapy. The ATS/IDSA guidelines do recommend obtaining a respiratory culture for all patients with suspected MRSA pneumonia [1]. These findings and guideline recommendations highlight the need for antimicrobial stewardship programs (ASPs) to optimize the utilization of the MRSA nasal PCR within a guideline-directed algorithm to reduce unnecessary anti-MRSA therapy.

There are numerous consequences to excessive antibiotic utilization, including increased rates of antibiotic resistance, adverse drug events (ADEs), and increased healthcare costs [15,16]. Furthermore, several distinctive drawbacks are associated with various anti-MRSA therapies. Potential adverse drug effects include nephrotoxicity and ototoxicity with vancomycin, bone marrow suppression and peripheral neuropathy with linezolid, and rhabdomyolysis or myopathies with daptomycin. The impact on antimicrobial consumption may be further highlighted when evaluating the WHO AWaRe classification of anti-MRSA antimicrobials. Vancomycin, daptomycin, linezolid, and ceftaroline are “reserve” antibiotics, while doxycycline and trimethoprim-sulfamethoxazole are considered “access” antibiotics [17]. Additionally, increased vancomycin use has contributed to rare, yet growing resistance in the forms of vancomycin-intermediate *Staphylococcus aureus* (VISA), heterogeneous VISA (hVISA), and vancomycin-resistant *Staphylococcus aureus* (VRSA) [18]. In addition to potential increased risk of ADEs, recent literature describes worse outcomes in patients who receive empiric anti-MRSA therapy for CAP. A 2019 retrospective cohort study of hospitalized CAP patients across the Veterans Affairs (VA) health care system demonstrated increased 30-day mortality with empiric anti-MRSA therapy compared to standard therapy alone [19]. The increased risk of mortality persisted, even in subgroups of patients who were admitted to an ICU, had clinical risk factors for MRSA infection, or tested positive on an MRSA nasal PCR. Additionally, a recent systematic review and meta-analysis by Carey et al. sought to estimate the effect of early empiric antimicrobials for MRSA on mortality. Based on their estimates, a baseline mortality of 30% and 10% prevalence of MRSA would be required to demonstrate a mortality benefit of empiric anti-MRSA therapy. Given that CAP has approximately a 5% prevalence of MRSA and 10% mortality for patients admitted to the general wards and 30% for patients admitted to the ICU, they conclude that empiric anti-MRSA therapy is unlikely to have a marked mortality benefit in the general population in the absence of MRSA risk factors [20]. This lack of projected efficacy benefit, paired with known increased risk of ADEs, healthcare costs, and antimicrobial resistance, further illustrate the need for appropriate empiric antimicrobial prescribing and prompt de-escalation of anti-MRSA antimicrobials in patients presenting with CAP.

Numerous studies evaluating the utility of MRSA nasal PCR de-escalation tools for pneumonia have demonstrated that negative PCR results reduce durations of anti-MRSA therapy [8,9,10,11,12,13,14]. However, there is a paucity of evidence assessing the impact of positive PCR results on durations of therapy. A retrospective cohort study by Acuna-Villaorduna et al. assessed the effect of MRSA nasal colonization on durations of vancomycin therapy at the VA Boston health care system, where all patients are screened for MRSA within 24 hours of admission [21]. In the cohort of patients with various infections, the median duration of vancomycin therapy was one day longer for patients with positive MRSA PCR results compared to those with negative results. However, a culture from the corresponding infection site was obtained for each patient, and there was no difference in duration of therapy in patients with negative MRSA cultures, regardless of PCR results. This differs from the present study, where respiratory cultures were only obtained in 46.2% of all patients, and MRSA PCR results may have played a more prominent role in the decision of whether to continue anti-MRSA therapy.

These results demonstrate the importance of sensitivity, specificity, NPV, and PPV when implementing antimicrobial stewardship tests within a guideline-directed diagnostic and treatment algorithm. Many rapid antimicrobial stewardship tests rely on high NPV but have low PPV, such as *Clostridioides difficile* PCR and (1,3)-β-d-glucan [22,23]. Due to time constraints, ASPs may be more inclined to focus on negative test results that may lead to high-yield interventions. However, these results indicate that ASPs should consider the potential ramifications of misinterpreted test results and opportunities for education and algorithm development to minimize any potential prolonged, negative impact of positive results.

The utilization of the MRSA nasal PCR may be optimized by emphasizing its NPV for rapid de-escalation of anti-MRSA therapy for the time between respiratory culture collection and subsequent bacteria growth. However, regardless of the PCR results, the presence or absence of MRSA growth on culture should be the final determinant of duration of anti-MRSA therapy in patients with pneumonia. Given the known consequences of inappropriate antimicrobial therapy, particularly anti-MRSA therapy, it is critical to limit antibiotic utilization to those patients who may truly benefit from them.

This study has several limitations, primarily related to the retrospective nature and small sample size. There was a reliance on documentation for diagnostic considerations and testing indications. Additionally, the cohort size was small, and a more robust sample size would increase the validity of these findings. However, our findings were consistent across several patient demographic categories, including the categorization of pneumonia and presenting severity of illness. Additionally, time from order of MRSA nasal PCR to results was not collected. Therefore, delays in obtaining nasal swabs may have impacted duration of therapy, though it is anticipated that this had a minimal impact, as evidenced by the similar overall durations of therapy and time from PCR result to discontinuation of anti-MRSA therapy.

## 5. Conclusions

Durations of anti-MRSA therapy were similar to definitive guideline-recommended pneumonia treatment courses in patients with a positive MRSA nasal PCR in the absence of MRSA growth on culture. These findings indicate that prescribers often interpret a positive MRSA nasal PCR as being equivalent to MRSA growth on cultures in the absence of microbiological sampling. While the MRSA nasal PCR is a valuable antimicrobial stewardship tool, these results highlight the need for education on the interpretation of positive tests to prevent excessive anti-MRSA therapy. ASPs that employ the use of MRSA nasal PCRs should seek to optimize the antimicrobial stewardship potential of these tests by incorporating them into a guideline-directed diagnostic and treatment algorithm for pneumonia. MRSA nasal PCRs should be used as a rapid de-escalation tool for negative results, and positive results should be interpreted in the context of patient-specific MRSA risk factors, clinical presentation, and respiratory or blood cultures, as recommended by guidelines, in patients presenting with pneumonia.

## Figures and Tables

**Table 1 pharmacy-11-00081-t001:** Patient demographics and culture data.

Characteristics	All Patients (*n* = 52)
Age in years, median (IQR)	61 (49.25–71.65)
Race, *n* (%)	
Caucasian	33 (63.5)
African American	17 (32.7)
Hispanic	2 (3.8)
Sex, *n* (%)	
Male	33 (63.5)
Female	19 (36.5)
Hospital length of stay in days, median (IQR)	10 (6–19)
Treatment location, *n* (%)	
Non-ICU	26 (50)
ICU	26 (50)
Indication for anti-MRSA therapy, *n* (%)	
**Respiratory infection**	**37 (71.2)**
CAP	28 (53.8)
HAP	8 (15.4)
VAP	1 (1.9)
**Non-respiratory infection**	**15 (30.8)**
Sepsis	4 (7.7)
Bacteremia	2 (3.8)
Skin and soft tissue infection	5 (9.6)
Urinary tract infection	3 (5.8)
Osteomyelitis	1 (1.9)
Respiratory culture obtained, *n* (%)	24 (46.2)
Culture specimen, *n* (%)	
Sputum	5 (20.8)
Bronchoalveolar lavage (BAL)	17 (70.8)
Tracheal aspirate	2 (8.3)

IQR = interquartile range.

**Table 2 pharmacy-11-00081-t002:** Anti-MRSA therapy duration.

Group	Total Duration of Anti-MRSA Therapy in Days, Median (IQR)	Duration of Anti-MRSA Therapy Following MRSA PCR Result in Days, Median (IQR)
All patients (*n* = 52)	5 (3–6)	4 (3–5.25)
Treatment location		
Non-ICU (*n* = 26)	4 (3–6.25)	4 (2.75–6)
ICU (*n* = 26)	5 (3–5.25)	4 (3–5)
Type of pneumonia		
CAP (*n* = 19)	4 (2.25–5)	4 (2.25–5)
HAP (*n* = 7)	7 (5.25–7)	6 (4.25–10.75)
VAP (*n* = 1)	3 (n/a)	3 (n/a)

IQR = interquartile range.

## Data Availability

The data presented in this study are available on request from the corresponding author. The data are not publicly available to ensure protection of patient privacy.

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
