# Peer review of "Evaluation of Anti-Methicillin-Resistant Staphylococcus aureus (MRSA) Prescribing Habits in Patients with a Positive MRSA Nasal Swab in the Absence of Positive Cultures"

_pharmacy, 2023, doi:10.3390/pharmacy11030081_

Round 1
Reviewer 1 Report
A very interesting study looking at appropriate duration of anti-MRSA therapy in positive nares MRSA PCR with negative cultures. It is a small study though with only 52 patients included.
Materials and Methods:
For inclusion criteria- did you review patients who were treated for pneumonia only or for any infection? Would clarify.
Would also include turn around time of MRSA PCR result. Do you have any data on time to order of MRSA nares PCR and results? Sometimes, there would be a delay in obtaining the swab which can affect time to discontinuation of anti-MRSA therapy.
Results:
I would include an algorithm of how you included/excluded patients.
Discussion:
There were 5 patients who had sputum samples sent for culture. Do you think the providers considered this an unreliable diagnostic tool (only representing oral flora and not coming from deeper structures)?
Author Response
A very interesting study looking at appropriate duration of anti-MRSA therapy in positive nares MRSA PCR with negative cultures. It is a small study though with only 52 patients included.
Thank you for your time, effort, and thoughtful comments. We hope that the revisions made based on your comments are to your liking and have improved the impact and quality of our manuscript.
Materials and Methods:
For inclusion criteria- did you review patients who were treated for pneumonia only or for any infection? Would clarify.
This study evaluated patients treated for any infection. The institutional protocol at our institution specifically provides guidance on MRSA nasal PCRs in patients with suspected pneumonia, but may be used for other indications at provider discretion. The methods section has been updated to clarify. This information can also be found in Table 1.
Would also include turn around time of MRSA PCR result. Do you have any data on time to order of MRSA nares PCR and results? Sometimes, there would be a delay in obtaining the swab which can affect time to discontinuation of anti-MRSA therapy.
Unfortunately, we do not have data on the turnaround time of the MRSA PCR collection. We have updated our limitations to address this while also noting that given the similarities between overall duration of therapy and durations of therapy after PCR result found in Table 2, we believe that delayed collection had a small impact on our findings.
Results:
I would include an algorithm of how you included/excluded patients.
We have updated the results to state our screening protocol and how many patients were excluded for each criteria.
Discussion:
There were 5 patients who had sputum samples sent for culture. Do you think the providers considered this an unreliable diagnostic tool (only representing oral flora and not coming from deeper structures)?
We have updated the discussion to provide more detail on CAP guideline recommendations on sputum culture (no recommendation for or against routine collection due to antimicrobial stewardship and diagnostic concerns). We also discussed the guideline recommendation in favor of collecting respiratory cultures in patients treated with anti-MRSA therapy. We hope this sufficiently addresses this important consideration and distinction.
Reviewer 2 Report
Staphylococcus aureus' s growing antibiotic resistance rises many problems in the heath care system. Each way to de-escalate this phenomenon is wellcomed. The study contributes with its rsults to the to prevention of excessive anti-MRSA therapy.
It is formulated in the title, which is in accordance with the content of the paper. Each work addressed to improving the antibiotic use is wellcome. It helps medical staff to think over the necessity of antibiotic prescription.
It referes to a specific location where the study was perfomed, it contributes to the medical staff’s education in this matter.
The conclusions are consistent with the evidence provided by the study.
The references are appropriate, no unnecessary reference is added.
Tables and figures are easy to understand.
The abstract, as the whole composition of the manuscript is well- understandable, scinetifically well organised, the tables are easily understood. The references are elevant and are cited adequatly.
It is interesing that the PCR results are positive even in cases where the bacteria cannot be cultivated. I wonder what is the reason of this phenomenon.
In all, congratulations to the authors.
Author Response
Staphylococcus aureus' s growing antibiotic resistance rises many problems in the heath care system. Each way to de-escalate this phenomenon is wellcomed. The study contributes with its rsults to the to prevention of excessive anti-MRSA therapy.
It is formulated in the title, which is in accordance with the content of the paper. Each work addressed to improving the antibiotic use is wellcome. It helps medical staff to think over the necessity of antibiotic prescription.
It referes to a specific location where the study was perfomed, it contributes to the medical staff’s education in this matter.
The conclusions are consistent with the evidence provided by the study.
The references are appropriate, no unnecessary reference is added.
Tables and figures are easy to understand.
The abstract, as the whole composition of the manuscript is well- understandable, scinetifically well organised, the tables are easily understood. The references are elevant and are cited adequatly.
It is interesing that the PCR results are positive even in cases where the bacteria cannot be cultivated. I wonder what is the reason of this phenomenon.
In all, congratulations to the authors.
Thank you for your time, effort, and thoughtful comments. We hope that this manuscript will further advance the field of antimicrobial stewardship and provide ASPs with additional information on education providers about appropriate interpretation of stewardship test results.
Reviewer 3 Report
Dear Authors,
Despite being a small study conducted in one single center, this is relevant because of the topic and the high importance of knowing the cause of inadequate prescription of antibiotics.
That said, in my opinion, it can't be accepted that a study submitted to a journal called Pharmacy, talking about antimicrobials, only refers to them as "anti-MRSA", without citing the name of any single molecule.
What is an anti-MRSA? Do you think all healthcare professionals understand the same by using this name? Are all "anti-MRSA" the same?
Certainly not! They belong to different chemical families, and their classification is different according to the WHO AWaRe categorization. Also, the potential consequences of their inappropriate use are different.
To be able to consider this research as a meaningful one, as a possible example for other settings and countries, it is paramount that there is a full description of the individual antibiotics used in this study, the recommended duration of use of each, dosages prescribed, an analysis antibiotic/indication of use, etc.
This is a drug utilization study; the message is clear: "While the MRSA nasal PCR is a valuable antimicrobial stewardship tool, these results highlight the need for education on the interpretation of positive tests to prevent excessive [use of certain antibiotics such as.....". Notwithstanding this, if the aim is to increase the education of prescribers and provide good guidance, prescribers must know which are involved antibiotics.
Author Response
Dear Authors,
Despite being a small study conducted in one single center, this is relevant because of the topic and the high importance of knowing the cause of inadequate prescription of antibiotics.
Thank you for your time, effort, and thoughtful comments. We hope that the revisions made based on your comments are to your liking and have improved the impact and quality of our manuscript
That said, in my opinion, it can't be accepted that a study submitted to a journal called Pharmacy, talking about antimicrobials, only refers to them as "anti-MRSA", without citing the name of any single molecule.
What is an anti-MRSA? Do you think all healthcare professionals understand the same by using this name? Are all "anti-MRSA" the same?
We have updated our methods section to specify which antimicrobial agents were included in the anti-MRSA evaluation.
Certainly not! They belong to different chemical families, and their classification is different according to the WHO AWaRe categorization. Also, the potential consequences of their inappropriate use are different.
We have clarified potential consequences of inappropriate antimicrobial therapy for several anti-MRSA agents used in the hospital setting (vancomycin, daptomycin, linezolid, etc) and included drug-specific side effects, costs, and resistance concerns (VISA, hVISA, and VRSA).
To be able to consider this research as a meaningful one, as a possible example for other settings and countries, it is paramount that there is a full description of the individual antibiotics used in this study, the recommended duration of use of each, dosages prescribed, an analysis antibiotic/indication of use, etc.
Thank you for this comment. We agree that it is important to understand the antimicrobials being utilized for anti-MRSA therapy. However, the focus of this study was on general trends of antimicrobial prescribing and de-escalation habits for the indication as a whole. As such, we chose to evaluate the appropriateness of anti-MRSA therapy as a group rather than individual agents within the context of specific diagnostic/stewardship test results. Patient specific factors such as renal function, presenting severity of illness, and would have prevented us from analyzing and providing generalizable data about the dosing of these agents in this review.
This is a drug utilization study; the message is clear: "While the MRSA nasal PCR is a valuable antimicrobial stewardship tool, these results highlight the need for education on the interpretation of positive tests to prevent excessive [use of certain antibiotics such as.....". Notwithstanding this, if the aim is to increase the education of prescribers and provide good guidance, prescribers must know which are involved antibiotics.
Thank you again for taking time to review this manuscript. We hope that the modifications made regarding specific antimicrobials evaluated and the consequences of inappropriate use specific to certain high-use qualifying agents improves the impact and quality of our manuscript.
Round 2
Reviewer 3 Report
Dear Authors,
Thank you for submitting this new and improved version of the manuscript.
Notwithstanding this, in my opinion, there is something still lacking. Certainly, in this new version, the individual antibiotics are cited ("Anti-MRSA agents included in this review include vancomycin, linezolid, daptomycin, ceftaroline, trimethoprim/sulfamethoxazole, and doxycycline). But, again, the use and misuse of these antibiotics do not have the same consequences. Vanco, linezolid, dapto and ceftar. are classified as "Reserve" antibiotics according to the WHO AWaRe classification; doxy and sulfa-trim are "Access".
The WHO AWaRe classification was established in 2017 (https://www.who.int/publications/i/item/2021-aware-classification). So, my suggestion is to try to use this classification for the analyses (Access anti-MRSA and Reserve anti-MRSA).
In case all the anti-MRSA antibiotics used in this study were Reserve, then this must be specified in the Methods/Results, and the Discussion and Conclusions because this has a direct impact on the messages to be included in stewardship actions.
Kind regards
Author Response
Thank you again for taking the time to review our manuscript and providing thoughtful feedback to further improve the quality of the submission. Please see our responses below to comments.
Dear Authors,
Thank you for submitting this new and improved version of the manuscript.
Notwithstanding this, in my opinion, there is something still lacking. Certainly, in this new version, the individual antibiotics are cited ("Anti-MRSA agents included in this review include vancomycin, linezolid, daptomycin, ceftaroline, trimethoprim/sulfamethoxazole, and doxycycline). But, again, the use and misuse of these antibiotics do not have the same consequences. Vanco, linezolid, dapto and ceftar. are classified as "Reserve" antibiotics according to the WHO AWaRe classification; doxy and sulfa-trim are "Access".
- We have added sentences to our discussion to state that the impact of these results on antimicrobial stewardship may be further highlighted based on antimicrobials used at a given institution and commented on the WHO AWaRe classification of each antimicrobial mentioned.
The WHO AWaRe classification was established in 2017 (https://www.who.int/publications/i/item/2021-aware-classification). So, my suggestion is to try to use this classification for the analyses (Access anti-MRSA and Reserve anti-MRSA).
- See point above. Thank you for providing this reference.
In case all the anti-MRSA antibiotics used in this study were Reserve, then this must be specified in the Methods/Results, and the Discussion and Conclusions because this has a direct impact on the messages to be included in stewardship actions.
- Thank you for this comment. Unfortunately, we only used antimicrobial administration as an inclusion criteria and did not track utilization of each antimicrobial. Therefore, specific antimicrobial utilization was outside of the scope of this study. The intention of this manuscript was to demonstrate the consequences of inappropriately interpreting a positive MRSA nasal PCR as equivalent to a positive culture. We are hoping to convey that by improving diagnostic accuracy and reducing rates of mis-diagnosis, we may promote improving antibiotic utilization among hospitalized patients regardless of anti-MRSA agent used by the institution.
Thank you again for taking the time to review our manuscript and for providing thoughtful feedback!